# Understanding the Impact of Walkability, Population Density, and Population Size on COVID-19 Spread: A Pilot Study of the Early Contagion in the United States

**DOI:** 10.3390/e23111512

**Published:** 2021-11-14

**Authors:** Fernando T. Lima, Nathan C. Brown, José P. Duarte

**Affiliations:** 1Stuckeman Center for Design Computing, The Pennsylvania State University, University Park, State College, PA 16802, USA; jxp400@psu.edu; 2Faculty of Architecture and Urbanism, Universidade Federal de Juiz de Fora, Juiz de Fora, MG 36036-900, Brazil; 3Department of Architectural Engineering, The Pennsylvania State University, University Park, State College, PA 16802, USA; ncb5048@psu.edu

**Keywords:** COVID-19, COVID-19 spread, walkability, population density, population size

## Abstract

The novel coronavirus disease 2019 (COVID-19) pandemic is an unprecedented global event that has been challenging governments, health systems, and communities worldwide. Available data from the first months indicated varying patterns of the spread of COVID-19 within American cities, when the spread was faster in high-density and walkable cities such as New York than in low-density and car-oriented cities such as Los Angeles. Subsequent containment efforts, underlying population characteristics, variants, and other factors likely affected the spread significantly. However, this work investigates the hypothesis that urban configuration and associated spatial use patterns directly impact how the disease spreads and infects a population. It follows work that has shown how the spatial configuration of urban spaces impacts the social behavior of people moving through those spaces. It addresses the first 60 days of contagion (before containment measures were widely adopted and had time to affect spread) in 93 urban counties in the United States, considering population size, population density, walkability, here evaluated through walkscore, an indicator that measures the density of amenities, and, therefore, opportunities for population mixing, and the number of confirmed cases and deaths. Our findings indicate correlations between walkability, population density, and COVID-19 spreading patterns but no clear correlation between population size and the number of cases or deaths per 100 k habitants. Although virus spread beyond these initial cases may provide additional data for analysis, this study is an initial step in understanding the relationship between COVID-19 and urban configuration.

## 1. Introduction

The novel coronavirus pandemic has significantly changed the way people interact with each other and with urban space. As there are no fully immunized cities yet, there is scientific consensus on the importance of adopting social distancing strategies to control the spread of COVID-19, especially in areas of high transmission rates or low vaccination rates [1,2,3,4]. Over time, many significant, increasing, and competing issues have arisen from implementing aggressive social distancing measures versus preserving socioeconomic activity. In this regard, there is a knowledge gap regarding how urban environments’ characteristics impact the spread of COVID-19 and infectious diseases in general. There is also a lack of, and an urgent demand for, data-driven approaches that can support decision-making related to these issues for different urban scenarios. The COVID-19 pandemic and all the complex data it generates point to the simple fact that contact leads to infection, and more face-to-face interactions are likely to increase transmission [1,3,4]. In this sense, cities are the stage where contact between people, and therefore infection, is more likely to occur. Although individuals and community behaviors such as how different groups choose to mask, stay home, get vaccinated and take other preventive measures also influence contagion spread, the available data indicates varying patterns of the spread of COVID-19 within American cities, especially in the first months when the contagion was faster in high-density and more walkable counties such as New York than in low-density and car-oriented counties such as Los Angeles (Figure 1). Thus, if an understanding is developed as to how urban features are correlated to different modes of social interaction and, consequently, to different patterns of COVID-19 spread, it will assist the identification of appropriate strategies to contain and mitigate the infection.

From a city-as-a-network (or as a network of networks) perspective, approached by several researchers [5,6,7,8,9], several localities made decisions to interrupt the transmission of COVID-19 through total lockdowns, which is akin to dismantling the entire network [9]. It is thus imperative to understand how such networks function and the effect of their links, hubs, clusters, gates, and so on. To obtain this understanding, it is crucial to investigate the role of urban features in infection dynamics.

This paper is part of a larger study aimed at the study of the correlation between urban form and Covid 19 propagation patterns. The larger study includes features related to population, spatial configuration, use patterns, and climate on the one hand, and features related to disease and control features on the other hand. The paper presents an exploratory study based on a set of regression analyses. The primary goal is to address an initial set of variables in a proof-of-concept experiment that seeks to preliminarily verify our hypothesis that certain urban features and associated spatial use patterns correlate with the spread of COVID-19. If an understanding is developed in this regard, it can partially explain and predict the future spread while identifying appropriate strategies to contain and mitigate the infection. The paper is focused on variables that are regulated in urban design tasks and for which accurate data is readily available. Namely, this paper thus verifies possible correlations between the COVID-19 spread in the United States and the following urban features: (i) walkability; (ii) population density; (iii) population size, and; (iv) the number of days in stay-at-home order for each location. To identify the influence of walkability, population density, and population size on the dissemination of COVID-19 in the United States, we have considered the 93 urban American counties (with population sizes from 200 k to 10 M) for which walkability data (Walk Score) was available and a 60-day time-lapse from the first case confirmation and death dates in each one of them according to USAFacts Database. This paper is structured in the following sequence: (i) short review on the COVID-19 pandemic and urban features (walkability, population density, and population size); (ii) description of the performed regression analyses; (iii) presentation and discussion of the results and; (vi) final remarks about the study and identification of intended future developments.

## 2. Background: Urban Features and Infectious Diseases Spread

According to [10], urban areas are the ground zero of the COVID-19 pandemic, responsible for 90 percent of reported cases before and during April 2020. The United States, in turn, was the number one country in the world in terms of COVID-19 cases and deaths as of 15 March 2021. From 21 January 2020 to 28 April 2020, 1,006,417 confirmed cases and 57,433 deaths were reported in the U.S.

There is a knowledge gap regarding how the configuration of urban environments impacts the spread of infectious diseases. As mentioned above, this work is an initial step in a broader research agenda that seeks to characterize and codify the relationship between urban features, social interaction patterns, and COVID-19 transmission, particularly in the context of American cities. This paper presents early results that, combined with others to come, can offer directives for designing alternative containment strategies for different urban contexts, from compact and walkable neighborhoods to sparse and car-oriented districts, and from the scale of ZIP code areas to counties. We intend to provide guidelines for interventions in existing cities to make them more resilient to infectious diseases and the future design of resilient cities. While the datasets and corresponding knowledge are specific to COVID-19 in the United States, our established methodology could be extended to predict the spread of future epidemics in other urban areas.

### 2.1. Walkability

According to several authors [11,12,13,14], a walkable urban area or an urban area that follows walkability’s principles considers pedestrians the highest priority, seeking greater urban life and promoting more socioeconomic interactions. According to this idea, walkability can be defined as a particular urban area’s capability to connect housing and amenities from several categories (e.g., retail, food, education, entertainment, and recreation) through distances that can be traveled within walking distance. This means more people walking, cycling, staying in public spaces, interacting, and exchanging information, as well as social and cultural opportunities. Thus, higher walkabilities increase the likelihood of people meeting and interacting due to the higher density of amenities. In this sense, we hypothesize that walkability acts as a proxy for several social interaction-related features and that places with greater walkability promote higher social interaction levels and, therefore, higher contagion rates of certain contagious diseases, such as COVID-19. In other words, we advocate that when walkability (as understood as a metric for the density of services) and population density increase, the likelihood of people meeting in places such as transport stations, public facilities, common entries, and elevators also increases. This effect might have been more apparent before other significant factors came into play. For example, masks were not recommended in the U.S. until 3 April 2020, and vaccines were not widely available until the following year. Accordingly, the work of [15] shows that, in Italy, the highest spread rates occurred in areas with commercial hubs, close to the highest populated cities, and the most industrial area. Their results indicate how human mobility can affect the epidemic, identifying particular situations in which the health authorities can promptly intervene to control the spread of the disease. Urban features in turn affect human mobility, and their influence is worth studying as well.

Several studies addressed ways of measuring the walkability of a particular location. For instance, the works of [16,17,18,19] consider the structure of street networks and their number of intersections, among others. However, in this research, we adopted the walk score index [20,21] for the following reasons: it is one of the most accessible walkability metrics (there are a lot of data available regarding the walk score of streets, neighborhoods, and cities in the U.S.); it was considered a reliable and valid measure of estimating walkable access to amenities; and walk score may be a convenient and inexpensive option for researchers interested in exploring the relationship between access to walkable amenities and health behaviors [22].

Walk score is an algorithmically obtained index for measuring an urban area’s walkability by assigning a score to a location based on its distance to various nearby services. The amenities considered by walk score can be divided into five categories: educational (e.g., schools), retail (e.g., grocery, drug, convenience, and bookstores), food (e.g., restaurants), recreational (e.g., parks and gyms), and entertainment (e.g., movie theaters). The algorithm calculates the distance to the closest of each of the five amenities categories. The results are normalized to a 0 to 100 scale, considering 0 as the lowest walkability (car dependent) and 100 as the highest (most walkable). For example, in relation to a particular locality, if one of the five amenities is within a 0.4 km (5 min walk) radius from the input location, then the maximum number of points, 100, is assigned to it. The number of points decreases as the distance increases to 1.6 km (30 min walk), and no points are awarded for locations amenities farther than 1.6 km. For instance, New York County and San Francisco County have high Walk Score indexes (88.3 and 87.4, respectively), while Chesapeake (Virginia) and Cumberland County (North Carolina) have extremely low walk score indexes (21 and 21.4, respectively).

### 2.2. Population Density

55% of the world’s population currently lives in urban areas, and this proportion is expected to increase to 68% by 2050 [23]. With people living in denser conditions, more interactions between individuals and disease transmission tend to occur more easily. As population density is an important urban feature that increases contact and, consequently, infection between people, several authors have studied the effect of population density on epidemic outbreaks in different contexts [24,25,26]. Still, the idea of high density of both population and buildings in urban areas is defended by several authors [12,27,28,29]. In the United States, population density is very heterogeneously distributed. For instance, New York County, Kings County, and Bronx County (all in New York) shelter, respectively, 71,876, 37,233, and 34,058 people per square mile. Washoe County (Nevada), Webb County (Texas), and San Bernardino Country (California) shelter, respectively, 74, 82, and 108 people per square mile.

### 2.3. Population Size

In addition to density and walkability, various socioeconomic interactions play an essential role in the dynamics of urban areas. As the overall size of a city is a critical aspect in defining social and economic life, it is also a relevant data point. Schläpfer et al. [30] advocate that different socioeconomic quantities increase superlinearly with city size and that this logic applies to almost all urban aspects, including the creation of new inventions and the prevalence of certain contagious diseases, for instance. At the same time, [31] state that the COVID-19 attack rate increases with city size and, in the absence of adequate controls, larger cities (and counties, as we assume) are expected to have more extensive epidemics than smaller ones. In the context of the United States and following this idea, Los Angeles County, California (10,039,107 inhabitants), Cook County, Illinois (5,150,233 inhabitants), and Harris County, Texas (4,713,325 inhabitants) would have the highest COVID-19 prevalence. Considering various hypotheses regarding relationships between different urban features, the population size should be compared to other factors.

### 2.4. Related Work

The recent COVID-19 pandemic stimulated the emergence of studies on the impact of population, spatial, and climatic features on the propagation of COVID-19 [32,33,34]. However, these studies are partial since they focus on just one or a few urban aspects. In addition, they are mainly focused on Chinese cities. On the other hand, Carozzi [35] states that density has affected the outbreak’s timing in American counties, with denser locations more likely to have a stronger outbreak. In turn, Oishi, Cha, and Schimmack [36] analyzed the role of walkability, wealth, and race in New York City, finding that walkability was negatively related to the number of COVID-19 cases and deaths. However, at the same time, the same authors identified that areas with a higher presence of certain ethnicities, median age, and occupants per room were more likely also to have higher COVID-19 cases and deaths. Dasgupta et al. [37] and Rocha et al. [38] address the role of socioeconomic vulnerability in the U.S. counties and Brazil, respectively. However, there is still a knowledge gap regarding how the characteristics of urban environments impact the spread of COVID-19 and infectious diseases in general. This work aims to contribute to bridging this gap by presenting an approach that seeks to find the relationship between urban features, social interaction patterns, and COVID-19 transmission, particularly in the context of American counties.

## 3. Method

### 3.1. Data

To verify whether there are correlations between certain urban features (walkability, population density, population size) and COVID-19 spreading patterns in urban areas, this work focus on county-level data, instead of city-level data, for two reasons: county-level data allow us to consider larger areas and more significant populations, but at a level of granularity that distinguishes between various townships (from big cities to surrounding small towns); and most of the available data on COVID-19 is organized at the county level. Thus, in addition to its practicality, we believe that addressing county-level data can provide more comprehensive information about the role of urban networks, enabling broader conclusions and increased freedom of analysis.

Instead of addressing a single and national timeline, considering the day of the first case in the United States as day one for all counties, we decided to study how the disease spread in diverse locations to identify how different urban features and associated urban patterns correlated. Our logic considers each county, regardless of their particularities, as a preliminary token to understand the whole country. To overcome potential bias in the timing of the disease’s onset across locations, we addressed the time-adjusted number of known cases and deaths per 100 k inhabitants in the studied counties. To this end, we considered two time-lapses for each county: 60 days after the first case (when addressing cases per 100 k hab) and 60 days after the first death (when addressing deaths per 100 k hab). The goal was to observe the longest time span possible and, at the same time, focus on spread in initial stages (when we assume that containment measures had less time to exert influence), allowing us to identify the effects of urban features more clearly. We used the software Minitab to run the analysis and the Grasshopper plugin for Rhinoceros was used for plotting some of the visualizations. Preliminary model fitting studies, carried out to identify the most suitable time interval, indicated the first 60 days as the best choice. After the first 60 days, both for cases and deaths per 100 k habitants, our R-squared adj values started to decrease significantly, as depicted in Figure 2.

When considering how to assess spread, death tolls are a more accurate indicator of COVID-19 prevalence since data on COVID-19 cases might be reported with error due to variation in local testing strategy and capacity [35,39]. However, we decided to address and compare both known cases and death tolls, as different aspects of health systems and underlying populations can influence the latter [40]. The number of known cases and death tolls were obtained from [41]. These data were combined with walk score [21], the number of days under a state-issued stay-at-home order [42], the population size of the counties [43] (total number of counties’ inhabitants), and the mean population density for each county [43] (total population/land area in sq miles). As Walk Score data were available on a city-level basis for 112 cities (from 200,000 to 10,000,000 inhabitants) and some cities were in the same county, and some counties were in the same city, it was necessary to aggregate data from the previous 112 cities into a final sample of 93 counties. Our sampling (Figure 3) allowed us to approach a total population of 115,791,837 people (35.27% of the U.S. population), 645,764 COVID-19 known cases, and 52,946 deaths (considering time adjustments).

### 3.2. Best Subsets Regression

The best subsets regression (BSR) is, together with forward stepwise and the lasso, the most popular methods for selecting and estimating parameters in a linear model. While the first two are understood as classical methods in statistics, the lasso is relatively more recent [43]. In a recent work, Hastie, Tibshirani, and Tibshirani (2020) [44] extensively addressed the potentialities and drawbacks of each of these methods, concluding that (1) neither BSR nor the lasso uniformly dominates the other; and (2) for a large proportion of the settings they considered, best subset and forward stepwise perform similarly, with BSR performing better in some specific situations. In turn, Bertsimas, King, and Mazumder (2016) [45] presented empirical comparisons of BSR with other popular variable selection procedures, including the lasso and forward stepwise selection. Their simulations suggested that BSR consistently outperformed the other methods in terms of prediction accuracy.

Thus, in this work, we adopt the (BSR) to test all possible combinations of the independent variables and select the best model according to goodness-of-fit criteria [46,47,48]. Since we are approaching a social and behavioral data analysis using multivariate regression, we are also interested in understanding the role of our potential predictors (independent variables) in the dynamics of disease spread. Thus, we adopted a correlation analysis, followed by a best subsets regression to determine which of our candidate independent variables (walk score, population density, population size, and the number of days in stay-at-home order) should be considered in our final regression model. This procedure was performed to build two regression models for comparison: one considering the number of cases per 100 k habitants 60 days after the first case in each county and the other considering the number of deaths per 100 k habitants 60 days after the first death in each county. We also performed an analysis considering population size data in its log-transformed state, but the model presented in this paper had a greater r. The goal was to use best subset regressions in the number of known cases per 100 k hab and deaths per 100 k hab against our set of independent variables to determine the most significant dependent and independent variables. Figure 4 illustrates our workflow for selecting the best regression model using the best subsets regression. Thus, we are approaching two conflicting considerations: minimizing the number of predictors to achieve a less expensive model and maximizing the model’s explanatory power. In addition to the regressions, we made a multivariable comparison between the counties with the higher and lower number of confirmed cases and deaths per 100 k habitants. These analyses generated preliminary findings that address the following questions: Which urban features matter most? Which can we ignore? How do urban features interact with each other?

## 4. Results

### 4.1. Correlation Analysis

In order to quantify the degree to which our independent variables are related and avoid biasing the models, we performed a correlation analysis considering all the addressed independent variables before running the best subsets regressions (Figure 5). Although walk score and population density present a correlation coefficient (r) of 0.582, indicating a moderate positive relationship (when 0.3 < r < 0.7), there is no strong relationship (when r ≥ 0.7) between any of the predictors. These correlation thresholds that supported our interpretation were applied together with graphical analysis and are broadly accepted guidelines for interpreting the correlation coefficient [49,50].

### 4.2. Best Subsets Regression

Following the check for correlation, our best subset regression analyses results (Table 1 and Table 2) show that all models that address the number of deaths per 100 k habitants presented higher values of adjusted R-sq, meaning that they fit the observed data better than the models that address the number of known cases per 100 k habitants. This analysis shows the two best models considering one independent variable, two independent variables, three independent variables, and the model containing all four independent variables. These models were compared to define which independent variables would be addressed in our final model.

When considering the first BSR alone (known cases per 100 k hab), the model addressing all four independent variables provided the highest R-Sq values and the lowest standard errors (S). On the other hand, when considering the second BSR alone (deaths per 100 k hab), the model addressing population density (PD), walk score (W.S.), and the number of days in stay-at-home order (D.O.) as independent variables provided the same R-Sq (adj) as the all-variables model, but with a lower value of standard error (S). Thus, when considering all subsets’ possibilities, we chose to build our final regression model considering the number of deaths per 100 k hab (60 days after the first day in each county) versus population density, walk score, and the number of days in stay-at-home order, since it provided us with the best R-Sq, S, and Mallows’ Cp values while addressing a smaller number of independent variables. R-sq informs the fitness of a model, S informs the standard error, and small Mallows’ Cp values indicate that the model has slight variance in estimating the accurate regression coefficients and predicting future responses.

### 4.3. Final Regression Model

Our analysis shows noteworthy correlations between walkability, population density, and the number of days at stay-at-home order with the number of deaths per 100 k hab, 60 days after the first case in each county (Table 3 and Table 4, and Figure 6). We came to the following findings after a normality test and a Box-Cox transformation of λ = 0.5 to our data. Our regression model provided an R-sq (adj) of 64.85% and a standard error (S) of 2.13467, which can be seen as very significant, especially if we consider that a set of non-measurable social behavior-related features such as how different groups choose to mask, stay home, and take other preventive measures also influence COVID-19 spread. The population density and walk score predictors presented *p*-values < 0.01, indicating solid evidence of statistical significance, while the number of stay-at-home days predictor presented a *p*-value < 0.05, indicating moderate evidence of statistical significance [51,52]. Overall, our Pareto chart of the standardized effects shows that walk score’s effect, population density’s effect, and days in order’s effect are more significant than the reference value for this model (1.987), meaning that these factors are statistically significant at the 0.05 level with the current model terms. Following these findings, our residual plot analyses (probability, fits, histogram, and order) validated the model.

Thus, our regression analyses positively correlated deaths per 100 k habitants and all independent variables. It means that as walk score, population density, and the number of days in stay-at-home order increases, these COVID-19 related numbers tend to be higher. Figure 7 depicts the evolution of cases and deaths per 100 k habitants through time, relating these numbers to each predictor and comparing the models for the number of cases and the number of deaths. Although it might seem controversial that the number of deaths increased with the number of days at home, our time-lapse sample, which intentionally addressed the initial stages of the spread, makes it reasonable to assume that places with higher disease spread adopted more robust measures as a reaction. Containment measures have a timing aspect that influences their performance. According to [53], the benefits of a lockdown are seen around 15–20 days before the peak of the epidemic, providing a limited window for public health decision-makers to mobilize and take full advantage of lockdown as an NPI.

### 4.4. Discussion

The COVID-19 pandemic and all of the complex data that it generates rely on a simple relationship: contact leads to infection. In this sense, cities are the stage on which contact between people and, therefore, the infection takes place. This preliminary study’s findings confirm our hypothesis that certain urban features (population density and walkability) correlate more with COVID-19 spread in the first days of the pandemic than other variables such as overall population size. Despite addressing an initial and limited set of predictor variables, we have found some important correlations (not a causal relationship, but an association to be further explored nonetheless). Considering our research scope, goals, and hypothesis (the impact of urban features on the disease spread), it is essential to highlight the importance of addressing the early stages of contagion to observe the trends before containment measures had a more significant influence.

Our results suggest a clear positive correlation between Walk Score and the number of deaths/100 k habitants, but it does not mean that the act of walking itself promotes higher contagion rates. Instead, it reinforces that places most likely to congregate amenities and promote encounters are potentially more contagious and require more effective containment actions.

## 5. Final Remarks: Limitations and Further Developments

### 5.1. Limitations of This Work

Despite achieving meaningful findings, the authors recognize that this study has some limitations. Disease spread during pandemics can be influenced by a host of social, economic, and behavioral factors in complex ways. In order to achieve more extensive results and increase our model’s fitness, it would be essential to address a broader sample of predictor variables, urban features, and urban scales, in addition to a more extensive time-lapse, weighted by other containment measures. Analyses at varying scales could, for example, address disparities within a single city across neighborhoods that are often starkly different despite their geographic proximity. Moreover, variables that capture heterogeneity across and within urban counties are both important. We also acknowledge that variables related to issues such as socioeconomic vulnerability, disproportionate spread in rural areas, where population sizes, population densities, and walkability indicators are small, population density expressed as the proportional population within a set of population density bands, and timing since the first case in the U.S. seem to be important and will be addressed in further developments.

On the other hand, this experiment provided an important basis for future work regarding the impact of urban features on COVID-19 (and similar contagious infections) spreading. Although some works have approached possible correlations of COVID-19 spread, population size, and population density in an American context [30,35], our research is, to the best of our knowledge, among the first to consider walkability as an important urban feature in this regard, since it is correlated with the interaction of people in urban environments as demonstrated in previous studies [20,21,22]. Our method assumes that the initial spread of the disease is less impacted by containment measures.

### 5.2. Future Work

In future stages of this research, we plan to address: (i) more urban features (e.g., mixed-use and floor area indexes, network density, volumetric compactness, containment measures, and so on); (ii) more urban scales (cities, zip codes, neighborhoods, and rural areas); (iii) a larger sample of time and cases (the timeline for the first 365 days in United States, for instance), (iv) socioeconomic, ethnic and racial indicators, (v) health-related indicators such as BMI, to identify physical activity levels in different places; (vi) urban mobility indicators; (vii) national and international connection measures and; (viii) data mining and machine learning techniques to retrieve, analyze, and model urban and infection data in different contexts. The expectation is that understanding how these features lead to different modes of social interaction and, consequently, to different dissemination patterns of COVID-19 will help identify appropriate strategies to contain and mitigate the infection and alternative healthcare policies.

More complex sets of features may also require the use of additional tools, such as data mining and machine learning techniques to retrieve, analyze, and model urban and infection data in different contexts. For example, the creation and analysis of an artificial neural network (ANN) might better capture the relationship between urban predictors and quantitative descriptors of COVID-19 spread. ANN is particularly useful when utilizing conventional engineering, or statistical approaches is not possible due to insufficient domain knowledge or the time and resource required makes it impractical.

Several researchers consider walkability and density key aspects of sustainable urbanism since they increase the potential of socioeconomic interactions and optimize energy performance, reducing pollution and resource consumption. However, our findings indicate that these urban features may directly correlate with a greater spread of COVID-19 in urban areas (at least in the United States scenario). In this context, urban planning may face some challenges in post-pandemic times to find answers to the following questions. How to balance the social, environmental, and economic need for such urban features with the need to provide more resilient and healthier cities? If a city can be conceived as a network, how can we ensure that it is flexible, efficient, and yet responsive to health? Which urban features have the greater impact on the spread of infectious diseases, and how can we manage them in this context? We believe that this work is one step in this direction.

## Figures and Tables

**Figure 1 entropy-23-01512-f001:**
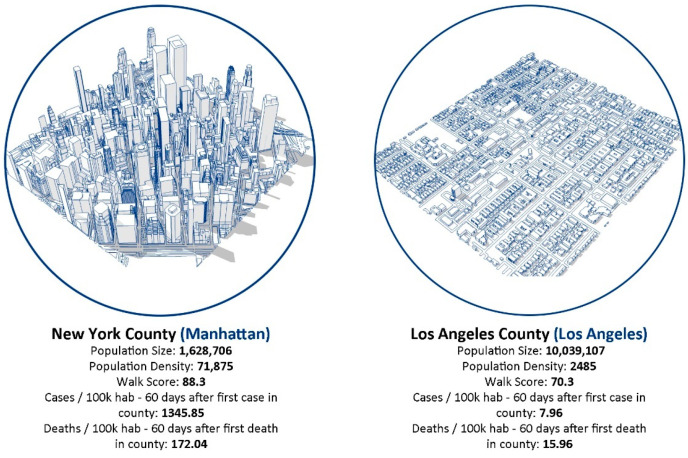
Basic urban features and COVID-19 dissemination data 60 days after the first case and the first death, according to USA Facts Database, in two counties addressed in this study: New York County and Los Angeles County.

**Figure 2 entropy-23-01512-f002:**
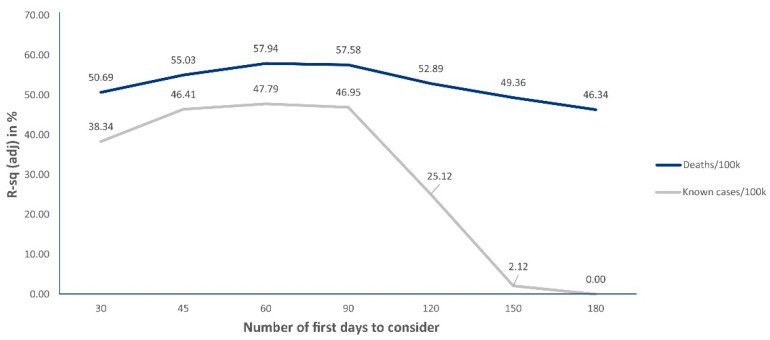
Preliminary linear model fitting results to determine the best time-lapse to address in regression analyses. The first 60 days performed better both for cases per 100 k hab and deaths per 100 k hab.

**Figure 3 entropy-23-01512-f003:**
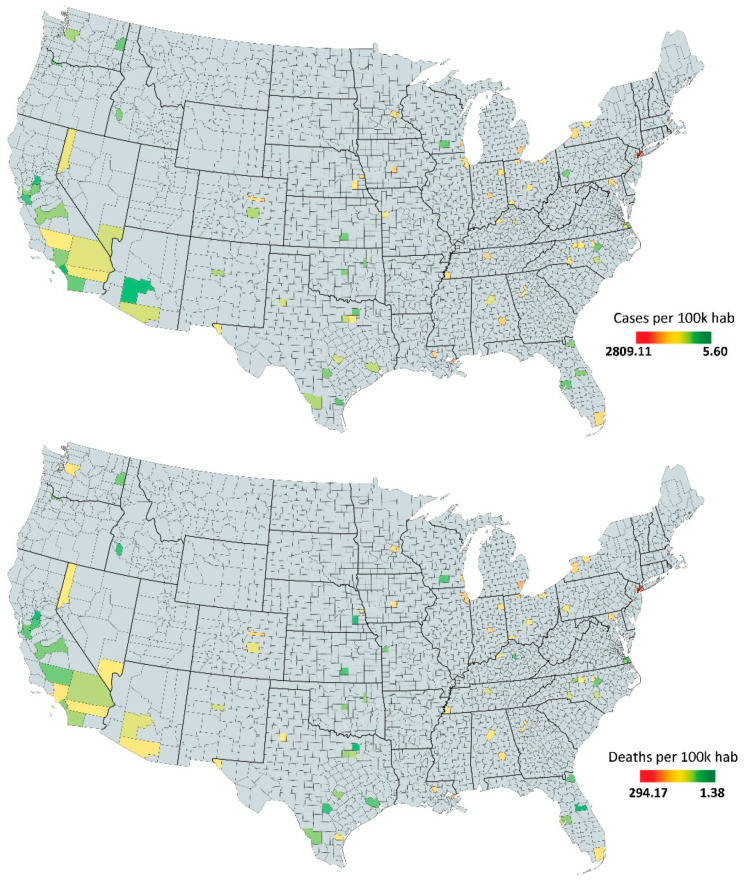
The 93 sampling counties of this study with a color map indicating the number of cases per 100 k inhabitants 60 days after the first case in each county (**above**) and the number of deaths per 100 k inhabitants 60 days after the first death in each county (**below**).

**Figure 4 entropy-23-01512-f004:**
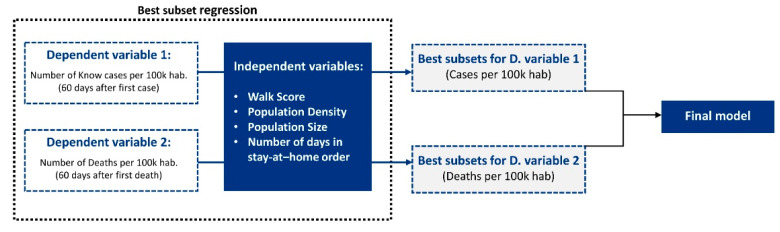
The workflow for building our final regression model: best subset regressions on the number of known cases per 100 k hab and deaths per 100 k hab were used against our set of independent variables to determine the most significant dependent and independent variables.

**Figure 5 entropy-23-01512-f005:**
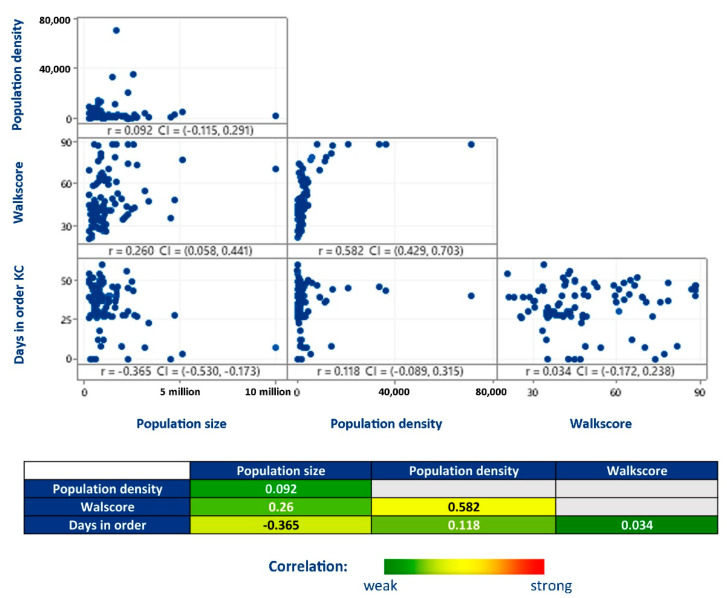
A correlation analysis considering all the addressed independent variables. Although walk score and population density present a moderate positive relationship (0.582), there is no strong relationship (≥0.7) between any of the predictors.

**Figure 6 entropy-23-01512-f006:**
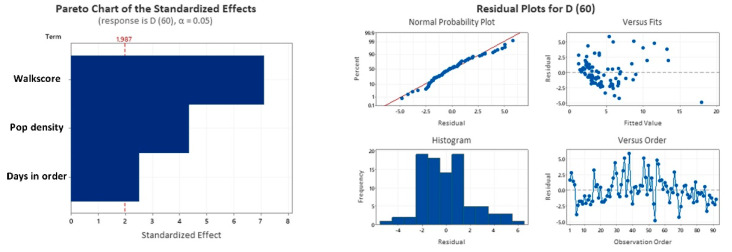
The Pareto chart of the standardized effects depicting the statistical significance of the addresses terms (**left**) and the residual plots for validating the model (**right**).

**Figure 7 entropy-23-01512-f007:**
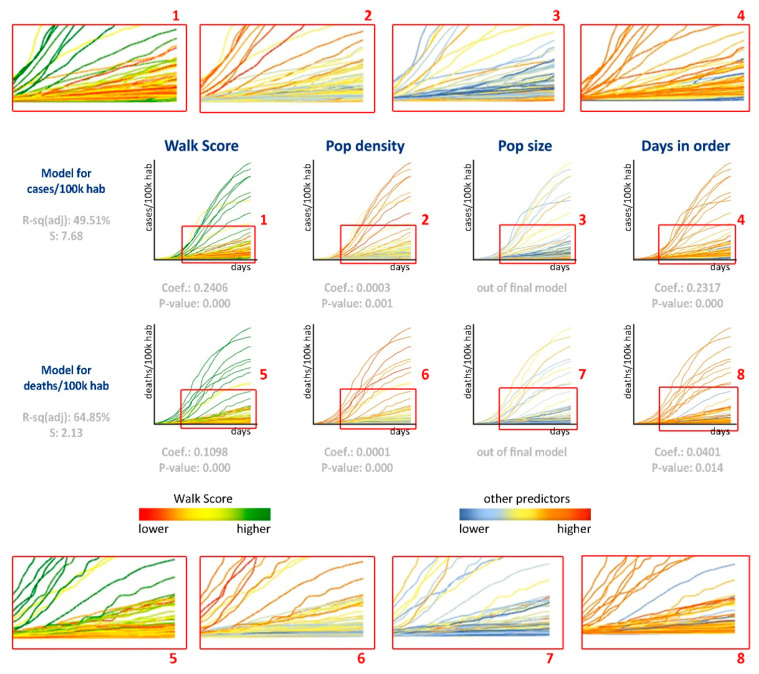
Cases per 100 k hab (**above**) and Deaths per 100 k hab (**below**) evolution in the 60 days after the first case (**above**) and death (**below**). Each line represents one of the analyzed counties. Different predictors weigh the data visualization.

**Table 1 entropy-23-01512-t001:** Best subset regression results considering the number of known cases per 100 k hab as the response. The model with all four independent variables (highlighted) provided the highest R-Sq (adj) and the lowest standard error (S).

Best Subset Regression Results 1—Response Is Know Cases per 100 k hab (after 60 Days from the First Case)					
Vars	R-Sq	R-Sq (adj)	R-Sq (pred)	Mallows Cp	S
1	39.2	38.5	33.7	29.1	462.24
1	34.8	34.1	0.0	37.4	478.43
2	46.9	45.7	41.0	16.1	434.22
2	46.9	45.7	10.9	16.2	434.36
3	53.0	51.4	20.1	6.5	411.03
3	51.7	50.0	16.9	9.0	416.65
4	54.8	52.7	21.8	5.0	405.32
**Vars**	**PD**	**WS**	**DO**	**PS**	
1		X			
1	X				
2		X	X		
2	X	X			
3	X	X	X		
3	X	X		X	
4	X	X	X	X	

**Table 2 entropy-23-01512-t002:** Best subset regression results considering the number of deaths per 100 k hab as the response. The model addressing population density (P.D.), walk score (W.S.), and the number of days in a stay-at-home order (D.O.) as independent variables (highlighted) provided the overall highest R-Sq (adj) and the lowest standard error (S).

Best Subset Regression Results 2—Response Is Deaths per 100 k hab (after 60 Days from the First Death)					
Vars	R-Sq	R-Sq (adj)	R-Sq (pred)	Mallows Cp	S
1	50.2	49.6	0.0	39.6	42.007
1	49.4	48.9	45.0	41.5	42.309
2	62.9	62.1	24.8	8.9	36.421
2	53.8	52.7	48.9	32.4	40.690
3	65.7	64.5	29.6	3.9	35.261
3	64.4	63.2	26.9	7.3	35.919
4	66.0	64.5	29.8	5.0	35.272
**Vars**	**PD**	**WS**	**DO**	**PS**	
1	X				
1		X			
2	X	X			
2		X	X		
3	X	X	X		
3	X	X		X	
4	X	X	X	X	

**Table 3 entropy-23-01512-t003:** Final model summary for transformed response (Box-Cox transformation λ = 0.5).

Regression Equation						
Deaths per 100 k hab^0.5= −2.672 + 0.000130 Population density + 0.1098 Walkscore + 0.0401 Days in order KC						
S	R-sq	R-sq(adj)	PRESS	R-sq(pred)	AICc	BIC
2.13467	66.01%	64.85%	631.932	46.44%	407.22	419.13

**Table 4 entropy-23-01512-t004:** Coefficients for the transformed response.

Term	Coef	S.E. Coef	95% CI	T-Value	*p*-Value
Constant	−2.672	0.918	(−4.496, −0.848)	−2.91	0.005
Population density	0.000130	0.000030	(0.000071, 0.000190)	4.33	0.000
Walkscore	0.1098	0.0155	(0.0791, 0.1406)	7.10	0.000
Days in order KC	0.0401	0.0160	(0.0084, 0.0718)	2.51	0.014

## Data Availability

Data is contained within the article.

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
