# Peer review of "Understanding the Impact of Walkability, Population Density, and Population Size on COVID-19 Spread: A Pilot Study of the Early Contagion in the United States"

_entropy, 2021, doi:10.3390/e23111512_

Round 1

Reviewer 1 Report

The authors present the results of regression analysis aimed to quantifying the effect of factors on covid cases and deaths. Among factors, walkscore is certainly innovative and not yet previously studied.
I only have a few suggestions for the authors:
1) The work states assumptions known in the literature. Although from a different geographical area, I think it is useful for the authors reading "A Municipality-Based Approach Using Commuting Census Data to Characterize the Vulnerability to Influenza-Like Epidemic: The COVID-19 Application in Italy", (Savini et al. 2020 ) since the population density is used for modeling the spatial diversity of the prevalence of covid at the starting of epidemics.
Please consider citing the work inside the introduction or at least at row 151.
2) The legend of figure 3 is the same as figure 2, please add the correct one.
3) This is the main issue. Considered factors are positively correlated with covid deaths. This is exepected for all the factors but number of days of restriction and, the explanation given by authors at rows 342 and following should be detailed and supported by data or literature.

Reviewer 2 Report

The study uses a regression approach to analyse the association between walkability, population density and population size and the spread of COVID-19 in US urban counties. The topic and approach are interesting and contribute to the debate on understanding the heterogeneity of COVID-19 spread in different areas. I have a few concerns with how the information is presented and with the interpretation of the findings.

Major issues:

The walkability score appears to be a measure of amenity distribution rather than interaction. By it's very nature, walkability captures a sense of how easily inhabitants can carry out trips by walking to avail of common amenities. Walking is not an activity associated with increased risk of COVID-19 transmission - if anything, it is an outdoor activity with low risk of transmission. What the variable does capture is a high density and distribution of locations where people go to and mix [in an indoor setting]. A number of the included amenities, such as retail, food and entertainment are private enterprises that set up in locations of high footfall either because people are passing through anyway (e.g. at a train station) or because people are drawn to the location for the purpose of using these amenities (e.g., shopping mall). That you can walk there seems irrelevant from a COVID-19 perspective - the issue is that they represent areas where people congregate. The higher the walkability, the more options people have for congregating because of higher density of amenities. In other words, the finding assumes that the walkability is the important feature when it is arguable that density of amenities is the significant issue. It is important that this gets teased out in the discussion.

The choice of time-lapse was based on an analysis of model fit based on different cut-point options. It is not entirely clear but it is implicit that the analysis was county-specific (i.e., 60 days after the first case/death in that county, rather than nationally). It would help to clarify that. However, the bigger issue is that the first case may have occurred at very different times across counties. It is plausible that inhabitants in late-affected counties may have taken a very different approach to risk mitigation than first-affected counties. In other words, seeing what happened in New York, for example, people in later affected counties may have been much more cautious about venturing out and mixing. As such, it would make sense to include an independent variable for timing since first case in the US.

Related to the above point is that certain cities, presumably those that are major international travel hubs, were more likely to have early and rapid spread of COVID-19. This was certainly the case in Europe, where northern Italy was the first hot spot due to its close links to China through the textile industry. As we are talking about an infectious disease with exponential growth - there is a big difference between sporadic cases arriving and plane loads of cases arriving. This could be addressed by, for example, including variables for people arriving in the city by plane [separately for international and national travel]. In the absence of this information, it is possible that your available variables are somehow characterising cities that are more attractive for inward travel. Tourist cities are often walkable cities.

The methods include an assumption that spread in the initial stages would be less impacted by containment measures. This again may depend on the stage of the epidemic generally rather than specifically in the county of interest. The type of measures and compliance with them will surely have an impact on spread, even in the early stages of the epidemic in any given county. I think this needs to be teased out as a limitation in the discussion section. Also, in contradiction to this, stay-at home orders are included as a covariate when surely they are a containment measure?

Population size was included as a dependent variable but found not to be a useful predictor. I was surprised that it was not included in its log-transformed state. Given the variability in city sizes in the US and the skewed nature of population size, surely it should have been transformed prior to inclusion and this could certainly change the interpretation of whether it is an important explanatory variable.

The discussion sets out a strong conclusion which, while technically correct, could easily be taken to imply much more than it actually shows. Based on an analysis with a highly limited set of predictor variables, you have found that they are correlated [note - not a causal relationship, just an association] with the spread of COVID-19. I would like to see that watered down a little, or at least strongly caveated to highlight that other potential explanatory variables/confounders were not analysed.

Why was no consideration given to including mobility measures (such as those available via Google and or Apple)? These would help you understand if walkability is correlated with mobility or if it is actually capturing something else. Other potentially important measures include BMI (which, if high, would suggest that people are not walking much), socio-economic status and census measures on preferred modes of transport.

Minor issues:

Population density is included as far as I can see as persons/square mile. This results in a very small coefficient which applies to an increase [or decrease] of 1 person per square mile. Given the kinds of magnitudes of difference that you would see, maybe rescaling to 10 or 100 persons/square mile would make the output easier to interpret. It would also help to state the mean values when describing the indicators in the methods section.

Could the authors please state the software used for the analysis?

Heterogeneity across and within urban counties are both important. Within county heterogeneity does not really seem to be captured. For example, population density could be expressed as proportion population within a set of population density bands. A county with medium population density could have an areas of high and low density - and using bands would enable you to evaluate that.

The approach to variable selection seems a remarkable over-complication given that there were only four included variables. Also, given that you know that you are likely to be missing important explanatory variables, it seems that more stress should be placed on including plausible variables rather than best-fitting variables. It would also help to list the kinds of additional variables that you would consider.

In figure 7, the plots show cases [and deaths] per 100k by days for each county. The plots are a nice addition. One thing that is not clear is whether the full spectrum of values are included in each plot. For example, the pop density plot for cases/100k only seems to show green through to red - I can't see any blues unless they are obscured by overlapping lines along the bottom of the plot. The pop size plots only seem to show blue to green/yellow. Is the spectrum set across the three predictors or separately for each predictor? For the walk score plots it is apparent that the full spectrum is represented.

There are a couple of grammatical errors or typos I came across:
 abstract: "or people moving from through those spaces"
 final remarks: "among the firsts to consider walkability"

Round 2

Reviewer 2 Report

The authors have revised the manuscript to address the comments from my earlier review. I appreciate the effort they have gone to and the detail they have provided in their responses. My only remaining concerns are as follows:

  1. The title should reflect that this is a pilot or exploratory study, so perhaps it should be amended to state ": a pilot study of the early contagion in the United States". It is really to acknowledge that this is not a final analysis, and the planned fuller analysis might well throw up different or even contradictory results.
  2. The analysis makes much of the influence of walkability on the spread of contagion. While the variable is constructed as a measure of walkability, I sincerely don't believe that is what is being captured for the purposes of contagion spread. Playing devil's advocate, one could create a headline from this paper that concludes that you are safer living in a city less suitable for walking in - "you are safer moving around by car" or "walking increases the spread of COVID-19!" One should not actually draw those conclusions here, as it is more likely that it is an association with amenity density and opportunities for population mixing. In a sense the measure has been repurposed here and that should be reflected in the language. While I think that has been dealt with in the discussion and methods, my worry would be that headlines often stem from the abstract - so maybe that would benefit from highlighting this nuance.
